# Advanced Modulation Format of Probabilistic Shaping Bit Loading for 450-nm GaN Laser Diode based Visible Light Communication

**DOI:** 10.3390/s20216143

**Published:** 2020-10-29

**Authors:** Guoqiang Li, Fangchen Hu, Peng Zou, Chaofan Wang, Gong-Ru Lin, Nan Chi

**Affiliations:** 1Key Laboratory for Information Science of Electromagnetic Wave (MoE), Fudan University, Shanghai 200433, China; 19210720066@fudan.edu.cn (G.L.); 18110720018@fudan.edu.cn (F.H.); 18110720058@fudan.edu.cn (P.Z.); 18210720152@fudan.edu.cn (C.W.); 2Graduate Institute of Photonics and Optoelectronics, Department of Electrical Engineering, National Taiwan University, Taipei 10617, Taiwan; grlin@ntu.edu.tw

**Keywords:** laser diode, visible light communication, probabilistic shaping, bit loading

## Abstract

Visible light communication is an emerging high-speed optical wireless communication technology that can be a candidate to alleviate pressure on conventional radio frequency-based technology. In this paper, for the first time, the advanced modulation format of probabilistic shaping (PS) bit loading is investigated in a high data rate visible light communication system based on a 450-nm Gallium Nitride laser diode. The characteristic of the system is discussed and PS bit loading discrete multi-tone modulation helps to raise the spectral efficiency and improve the system performance. Higher entropy can be achieved in the same signal-to-noise ratio (SNR) and modulation bandwidth limitation, comparing to bit and power loading. With PS bit loading, an available information rate (AIR) of 10.23 Gbps is successfully achieved at the signal bandwidth of 1.5 GHz in a 1.2 m free space transmission with normalized generalized mutual information above 0.92. And higher AIR can be anticipated with an entropy-loading strategy that fixes the channel characteristic. Experimental results validate that a PS bit loading scheme has the potential to increase the system capacity.

## 1. Introduction

Due to the great demand for higher data rate transmission and the over-crowdedness of the bandwidth of conventional radio frequency (RF) based communication technology, the use of visible light communication (VLC) as an alternative technology has aroused considerable interest from researchers. With the wide unlicensed spectrum, VLC has the potential to transmit high-speed data without electromagnetic interference [1,2,3,4]. Combining illumination and communication, VLC has many application scenarios, such as indoor positioning, underwater communication, and intelligent vehicles system. Light-emitting diodes (LED) and laser diodes (LD) are two kinds of transmitters in a VLC system. In a previous report, with a single-color LED, more than a 3 Gbps data rate was successfully achieved [5] and more than a 20 Gbps VLC system based on a multichromatic LED array chip was experimentally demonstrated utilizing wavelength-division-multiplexed technology [6]. Micro-LED was able to accomplish around a 10 Gbps data transmission rate based on a higher bandwidth [7]. However, due to the limited modulation bandwidth of LEDs at 10 s–100 s MHz and the efficiency droop problem for micro-LED [8], realizing a much higher data rate (gigabit class range) is still a challenge.

In contrast to LED, LD has a much higher modulation bandwidth and the highest electrical-to-optical conversion efficiency [9]. What’s more, LD owns the advantages including light beam convergence and high optical power. Therefore, LD may be a better candidate for front-end transmitters in a VLC system to realize a high Gbps data rate. In [10], Chun et al. realized a data rate of around 6 Gbps by utilizing orthogonal frequency-division multiplexing (OFDM) with a fixed-rate and bit loading scheme. Lin’s group demonstrated 9 Gbps error-free data transmission in a 450-nm GaN LD-based VLC system with the bandwidth of 1.5 GHz [11] and they further extended the bandwidth to 3.2 GHz through an impedance matched hardware circuit, achieving a data rate of 14 Gbps [12]. A higher data rate of 15 Gbps was accomplished by Viola et al., with adaptive bit loading extending the effective bandwidth to 2.5 GHz [13], and a 16.6 Gbps data transmission was realized in [14] by using bit-power loading and post equalization. LD emerges to be a promising light source for high-speed VLC applications [15].

The aforementioned work done on LD utilized bit loading or bit and power loading to improve system performance. By reallocating the available power and distributing the total number of bits over subcarriers, bit and power loading helps to raise the spectral efficiency [16]. However, the allocated bit can only be an integer and there still exists a capacity gap to the Shannon limit. As an emerging cutting-edge constellation shaping scheme, probabilistic shaping (PS) can provide additional shaping gain by changing the probability of each constellation point to make a Gaussian-like constellation [17,18]. PS can be combined with bit loading to further eliminate the gap to the Shannon capacity.

In this paper, for the first time, PS bit loading discrete multi-tone (DMT) modulation is investigated in a high data rate VLC system based on a 450-nm GaN LD. Utilizing the proposed entropy-loading strategy, higher entropy can be achieved compared to bit and power loading. The characteristic of the system is also discussed in detail. With PS bit loading, an achievable information rate (AIR) of 10.23 Gbps was achieved over 1.2 m in free space. The normalized generalized mutual information (NGMI) among all subcarriers were above 0.92, showing that there exist margins in the system. Experimental results show the potential of PS bit loading to increase the system capacity.

The following sections are organized as follows: the principle of PS bit loading is introduced in Section 2. Section 3 shows the experimental setup of the system. The system performance will be discussed in Section 4. In Section 5, we will draw the conclusion.

## 2. Principles

The block diagram of the probabilistic shaping bit loading scheme is shown in Figure 1. First of all, channel estimation is done by sending a quadrature phase-shift keying (QPSK) DMT signal as the training sequence. DMT is a common form of multicarrier modulation. The principle of DMT modulation can refer to [19]. In the experiment, we assume the transmission link between transmitter and receiver is relatively stable and the change of distance will only influence the signal-to-noise ratio (SNR) of the received signal. So, the impact of the distance link will not be considered in detail. SNR for each DMT subcarrier can be calculated through the error vector magnitude (EVM) of the received signal [20] as
(1)$$SNR=EVM_{\mathrm{RMS}}^{2},$$
the $EVM_{\mathrm{RMS}}$ is defined as
(2)$$EVM_{\mathrm{RMS}}={\left[\frac{\frac{1}{T}\sum_{t=1}^{T}{\left|I_{t}-I_{0,t}\right|}^{2}+{\left|Q_{t}-Q_{0,t}\right|}^{2}}{\frac{1}{N}\sum_{n=1}^{N}\left[{\left(I_{0,n}\right)}^{2}+{\left(Q_{0,n}\right)}^{2}\right]}\right]}^{\frac{1}{2}},$$
where $I_{t}$, $Q_{t}$ denote the in-phase (I) and quadrature (Q) components for the t-th received symbol, $I_{0,t}$, $Q_{0,t}$ denote the I and Q components for the t-th transmitted symbol, T is the total number of the received symbols, and N is the number of the unique symbols.

Then, the PS bit loading signal is generated according to the SNR. The source entropy of each subcarrier is set to the Shannon capacity as
(3)$$H=\log_{2}(1+SNR),$$

For quadrature amplitude modulation (QAM) constellation, the Gaussian-like constellation satisfies the Maxwell–Boltzmann (MB) distribution [17] and the MB distribution for PS M-QAM is expressed as
(4)$$P_{X}(x_{i})=e^{-v{\left|x_{i}\right|}^{2}}/\sum_{j=0}^{M-1}e^{-v{\left|x_{j}\right|}^{2}},$$
where v is adjusted to match the source entropy and the set $\left\{x_{0},x_{1},\ldots ,x_{M-1}\right\}$ contains the complex value of the M-QAM constellation. In this paper, the Constant Composition Distribution Matching (CCDM) algorithm [21] is utilized to realize probabilistic constellation shaping, through which the random input bits can be mapped to a probabilistically-shaped output constellation with specific source entropy. And the famous probabilistic amplitude shaping (PAS) scheme with MB distribution is employed to realize the PS system. The PAS scheme can be referred to [22]. Then the FEC encoder [23] is followed by the CCDM and it should be noted that the FEC encoder will not change the probability of the symbol.

Theoretically, the PS M-QAM constellation of a subcarrier is generated by employing a CCDM separately. So, the complexity of the proposed modulation format is relatively high considering the implementation of a large amount of CCDM for the overall subcarriers. However, in practical application, the complexity can be reduced by sharing the CCDM [24]. Through adaptively partitioning and precoding the subcarriers, the SNRs of several subcarriers are averaged and the source entropy loaded on these subcarriers can be the same, which means that the use of a CCDM is able to finish the data generation for several subcarriers. Besides, with the rapid development of chip integration and computing power, the complexity of the proposed modulation format will not restrict its realization.

In the PS system, GMI and NGMI are often used to measure the system performance for convenience [25]. And NGMI has been proved to be suitable for the prediction of the bit error rate of the forward error correction (FEC) after soft decision bit-metric decoding [26]. The GMI can be calculated as
(5)$$GMI\approx \frac{1}{N}\sum_{i=1}^{N}\left[-P_{X}(x_{i})\log_{2}P_{X}(x_{i})\right]-\frac{1}{N}\sum_{i=1}^{N}\sum_{j=1}^{m}\log_{2}(1+e^{{\left(-1\right)}^{b_{i,j}}\Lambda_{i,j}}),$$
where $b_{i,j}$ are the transmitting bits of i-th symbol and $m=\log_{2}M$ [25]. The NGMI can be expressed as
(6)NGMI=1−(H−GMI)/m,

In this paper, we set NGMI threshold as 0.92 for the FEC code rate 9/10 to realize error-free decoding [26]. So, the AIR can be calculated as
(7)AIR=(H−m(1−Rc))BW,
where Rc denotes the code rate and BW is the signal bandwidth.

## 3. Experimental Setup

Figure 2 shows the experimental setup of a laser diode-based VLC system utilizing advanced modulation format of PS bit loading. In the training phase, a QPSK DMT signal was used as the training sequence to test the VLC channel and the SNR table was built by channel estimation. The process of the DMT modulation can refer to [27]. Based on the specific SNR table, the source entropy for different DMT subcarriers can be determined through the entropy-loading strategy. In the testing phase, I- and Q-path PS-pulse amplitude modulation 32 (PS-PAM-32) signal with an optimal MB distribution were generated by a CCDM and an FEC encoder. And the PS QAM 1024 signal was generated by the corresponding PS-PAM-32 symbols. Then, the PS bit loading signal was generated after DMT modulation.

At the transmitter side, the analog electrical PS bit loading signal output from the arbitrary wave generator (AWG, Agilent, Beijing, China, M8190 A, 12 GSa/s) was amplified by an electrical amplifier (EA, Mini-Circuits, New York, United States, ZHL-2-8S+, 10–1000 MHz). Then the signal was coupled with DC current and fed into a GaN LD (OSRAM, Munich, Germany, PL 450B) with a nominal emission wavelength of 450 nm.

As we know, LD will continue to heat up, which affects the reliability and stability of the system. Keeping the temperature stable for better performance, the blue LD was specifically packed in a copper mount with a thermos-electric cooler (TE cooler). With the help of such a module design, the LD keeps high modulation efficiency and good power stability even driven at relatively large DC currents. In the experiment, we found that the LD performs better when the temperature is set to 24 °C. So, in the following experiment, the temperature was maintained at 24 °C with the use of the TE cooler. The divergent light of the LD was gathered by a lens and transmitted into free space.

At the receiver side, an avalanche photodiode (APD module, Hamamatsu Photonics, Hamamatsu, Japan, C5658), with a cut off frequency of 1 GHz, was located at the distance of 1.2 m from the LD to detect and convert the light into an electrical signal. In front of the APD, a neutral density filter (ND filter, Daheng Optics, Beijing, China, GCO-0702M) was employed to adaptively attenuate the incident light, avoiding the saturation of the APD. It should be noted that about 9.6 dB insertion loss of optical power will be induced in the light path. Hence longer transmission distance can be achieved without the ND filter. But if the transmission distance is too long, the power of the received signal will be decreased which results in the degradation of the system performance. The signal detected from the APD was fed to and recorded on a digital storage oscilloscope (OSC, Agilent, Beijing, China, MSO9254A) for further offline signal processing. In the experiment, AIR was used as a metric to assess the system performance and it can be calculated through GMI according to Equation (5). Furthermore, NGMI performance was also evaluated (Equation (6)) as an FEC threshold for PS bit loading DMT signal.

## 4. Results and Discussion

To study the performance of the advanced PS bit loading scheme, we first evaluate the characteristic of the system. Figure 3 shows the luminance-voltage-current (L-V-I) characteristics of the LD which were experimentally obtained at 24 °C. The threshold current of the LD is 25 mA. And a large linear operating area for the LD can be verified from Figure 3.

Before transmitting the PS bit loading DMT signal, the frequency response was often characterized as a figure of merit to evaluate the allowable direct modulation bandwidth of the whole system. Figure 4 shows the S21 transfer response of the system at different driven currents, which was measured by a network analyzer. The decrease throughout in intensity at the high-frequency region is mainly caused by the bandwidth limitation of the amplifier and the APD. Theoretically, raising the driven current makes the LD up-shift its relaxation oscillation frequency which results in the extension of the modulation bandwidth [11]. However, from Figure 4, we find that the high-frequency response was declined as the driven current increased. Additionally, it was caused by the saturation of the APD. Although an absorptive ND filter was utilized to attenuate the incident light on the APD, the light attenuation was not enough when the output power was relatively large at a high driven current. As a consequence, the modulation bandwidth (–10 dB) of the system is about 1 GHz due to these constraints.

To further discuss the transmission performance of the system, we searched for the optimal operating point by sending PS bit loading DMT data at the bandwidth of 1.5 GHz. Firstly, SNR estimation was done at different driven currents, and Figure 5a shows the SNR versus subcarrier index. At this time, signal Vpp was fixed at 200 mV. At low driven current region, the output power of the LD was limited, and the system was constrained by the whole SNR. Increasing the driven current from 37 to 57 mA increased the SNR among all subcarriers. However, the overly driven current of larger than 57 mA further increased light intensity and made the APD saturated, which resulted in the reduction of the SNR. And it should be noted that the enlarged driven current declines the frequency response at the low-frequency region, which may also deteriorate the system performance. The SNR distribution of subcarriers corresponds to the frequency response shown in Figure 4.

Figure 5b shows the source entropy versus subcarrier index at driven current at 37 mA and 57 mA. In a sense, the distribution of the SNR reflects the source entropy loading of subcarriers, according to the entropy-loading strategy. Table 1 shows the GMI at different driven currents. The highest GMI can be obtained at a driven current of 57 mA. And GMI at 37 mA was lower than that at 57 mA even with a larger modulation bandwidth due to the SNR limitation.

At the optimal driven current of 57 mA, we measured the SNR of all subcarriers at different signal Vpp, as represented in Figure 6. A similar tendency can be found just as is shown in Figure 5a. Increasing the signal Vpp raised the SNR of the received signal, thus improved the system throughput. This indicates the linear operating area of the LD and the output power varies linearly with the signal Vpp. Table 2 shows the GMI at different signal Vpp. At the signal Vpp of 240 mV, we achieved the highest GMI of 7.03 bits/symbol. When the signal Vpp exceeds 240 mV, nonlinear distortion will be introduced, and it causes performance loss for the system. In the following experiments, the operating point of the system was selected as driven current at 57 mA and signal Vpp at 240 mV.

As mentioned before, the modulation bandwidth severely decreases at a high-frequency region due to the bandwidth limitation of the system devices. Under the optimal operating point, we compared the system performance at different signal modulation bandwidths. Figure 7a shows the SNR in every DMT subcarrier. Increasing the signal bandwidth enlarges the attenuation at a high-frequency region. And the whole SNR will be decreased due to power allocation under the total power budget. Figure 7b–f shows the electrical spectra of the transmitted signal (Tx) and received signal (Rx) at different bandwidths, from which we can understand the system frequency response. The electrical spectra begin to attenuate with a greater negative slope when the signal bandwidth is larger than 1 GHz, which represents the limited system bandwidth.

Under the optimal operating point of the system, source entropy and GMI versus subcarrier index at different signal bandwidths are illustrated in Figure 8a–e. SNR represents the channel capacity to a certain extent and the source entropy obviously follows the trend of channel capacity according to Equation (3). As the gray region shown in the figure, the gap between the source entropy and the GMI is narrow, which means that PS bit loading scheme brings great shaping gain for the system and helps to raise the spectral efficiency. And source entropy adjustment method can be utilized to further improve the system performance. We also give the QAM order that the bit and power loading can achieve in the same SNR limitation. The QAM order was calculated under the 7% FEC threshold. Compared to PS bit loading, the QAM order of bit and power loading can only be an integer, and the entropy in each subcarrier is much lower than with PS bit loading. Therefore, the spectral efficiency of PS bit loading is higher than that of bit and power loading. With PS bit loading, the capacity of the system can approach the Shannon limit, providing higher data rates under the same power budget. Besides, the source entropy of the subcarriers can be adjusted for flexible rate adaption according to different channel conditions, making the transmission link more stable and reliable.

Figure 8f shows the NGMI versus subcarrier index under the optimal operating point at the signal bandwidth of 1.5 GHz. The minimum NGMI is 0.932, located in the 31th subcarrier. And NGMI in all subcarriers are strictly above 0.92, revealing that with 8.7% FEC overhead, error-free post-FEC results can be produced [25]. It should be noted that there is still a gap between the tested NGMI and the NGMI threshold of 0.92, so there exist margins in the system. Figure 9 illustrates the constellation diagrams of the received signal at the signal bandwidth of 1.5 GHz with different entropy from 9 to 4 bits/symbol.

The AIR versus different signal modulation bandwidths was measured, as shown in Table 3. When the bandwidth was smaller than 1.5 GHz, the AIR increased as the bandwidth enlarged. At this time, the system was bandwidth-constrained so the bandwidth extension improved the system performance. However, the AIR decreased when the bandwidth enlarged overly (larger than 1.5 GHz). Because the system was SNR constrained and the high-frequency response was severely attenuated. The results indicate that the highest AIR of 10.23 Gbps can be achieved at the bandwidth of 1.5 GHz through PS bit loading. Besides, we also calculate the minimum NGMI among all subcarriers at different bandwidth, all of which are above 0.92, showing that higher AIR can be achieved in the existing system.

## 5. Conclusions

This paper demonstrated a high-speed VLC system employing a 450-nm GaN LD. Due to the characteristic of the system devices, the modulation bandwidth (–10dB) of the system is about 1 GHz. PS bit loading DMT modulation was utilized to raise the spectral efficiency at such limited bandwidth. As far as we know, this is the first time that a PS bit loading scheme is applied to an LD-based VLC system. Compared to bit and power loading, higher entropy can be achieved in the same SNR and modulation bandwidth limitation. Experimental results validate the potential of a PS bit loading scheme for AIR improvement. With PS bit loading, we successfully achieved an AIR of 10.23 Gbps at the signal bandwidth of 1.5 GHz. The free space transmission distance was set at 1.2 m and longer distance may be anticipated without about 9.6 dB insertion loss of optical power induced by the absorptive ND filter. The NGMI among all subcarriers were strictly above 0.92, revealing the error-free transmission with 8.7% FEC overhead. The gap between the tested NGMI and the NGMI threshold shows there exist margins in the system. So, a more suitable entropy-loading strategy can be studied in the future to further improve system performance.

## Figures and Tables

**Figure 1 sensors-20-06143-f001:**
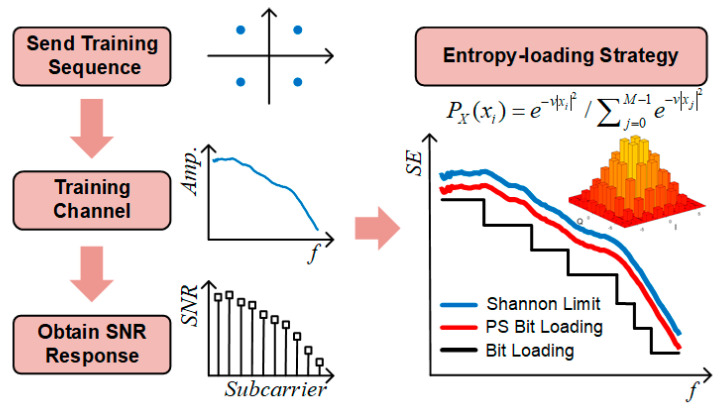
Block diagram of probabilistic shaping bit loading scheme.

**Figure 2 sensors-20-06143-f002:**
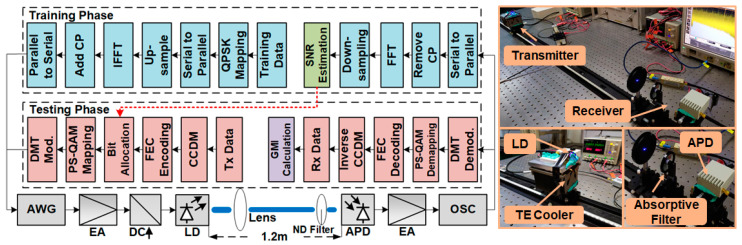
The experimental setup of a laser diode-based VLC system.

**Figure 3 sensors-20-06143-f003:**
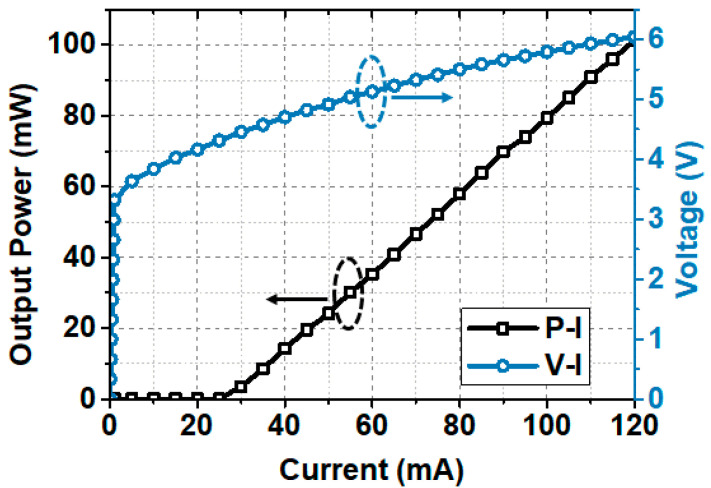
The L-V-I characteristics of the LD at 24 °C.

**Figure 4 sensors-20-06143-f004:**
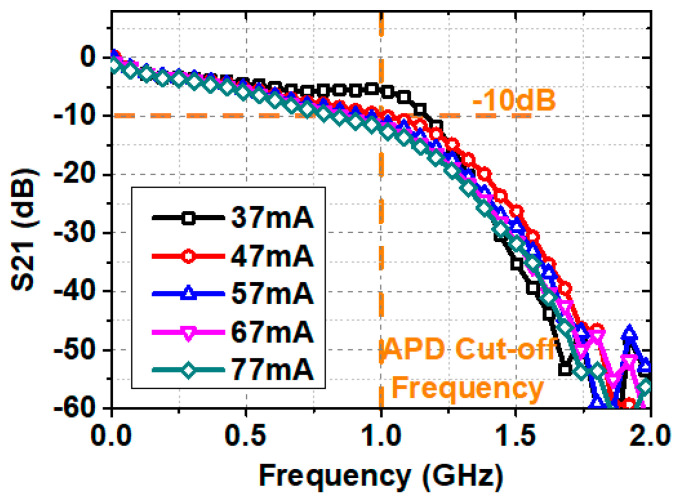
The S21 plot at different driven currents.

**Figure 5 sensors-20-06143-f005:**
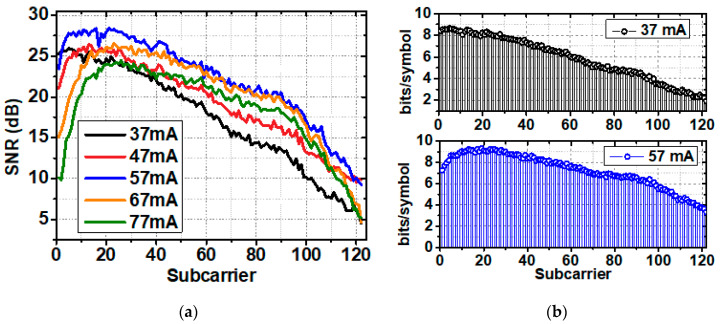
(**a**) SNR versus subcarrier index at different driven currents; (**b**) Source entropy versus subcarrier index at driven current of 37 mA and 57 mA.

**Figure 6 sensors-20-06143-f006:**
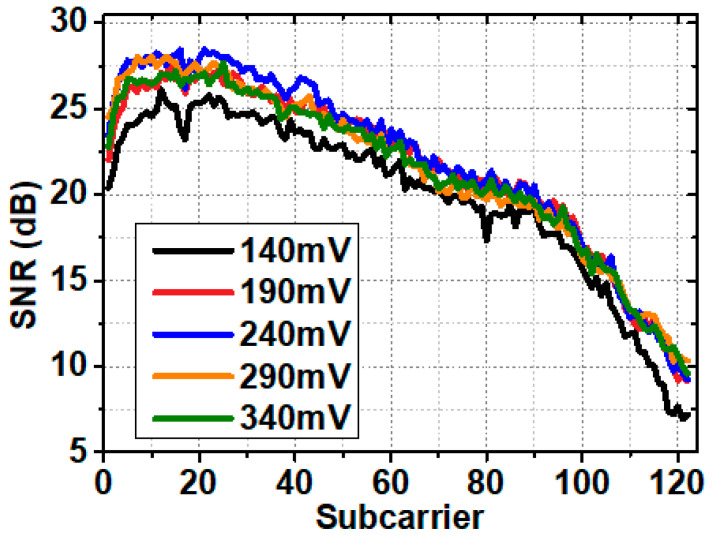
SNR versus subcarrier index at different signal Vpp.

**Figure 7 sensors-20-06143-f007:**
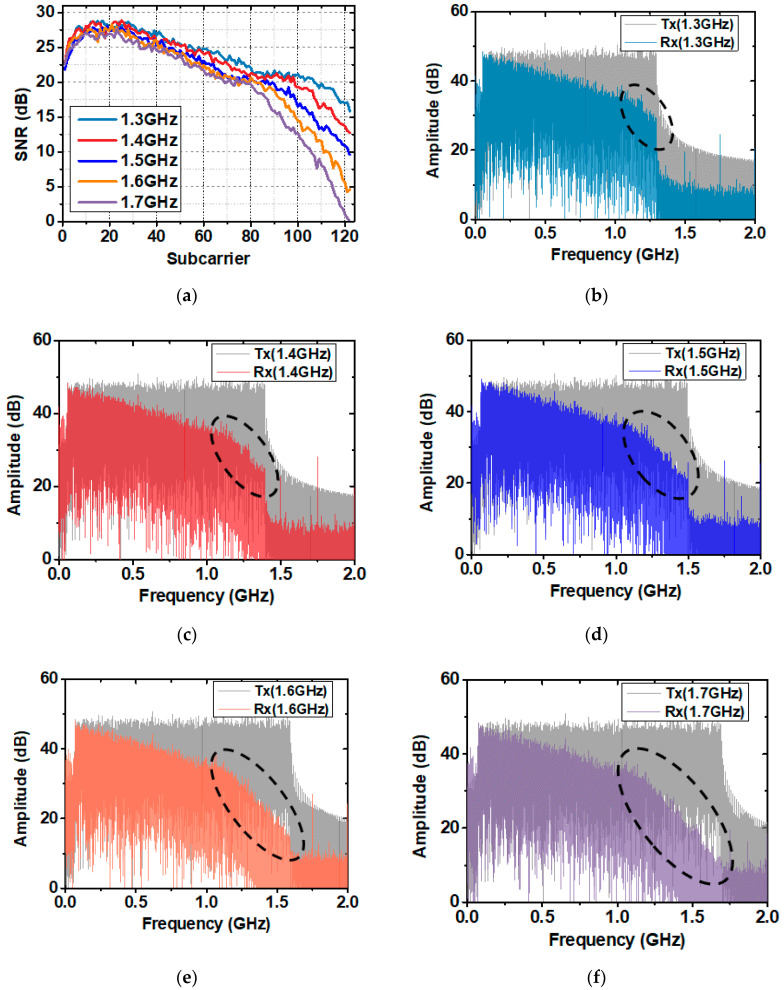
(**a**) SNR versus subcarrier at different signal bandwidths; (**b**–**f**) The electrical spectra of transmitted signal (Tx) and received signal (Rx) at different signal bandwidths.

**Figure 8 sensors-20-06143-f008:**
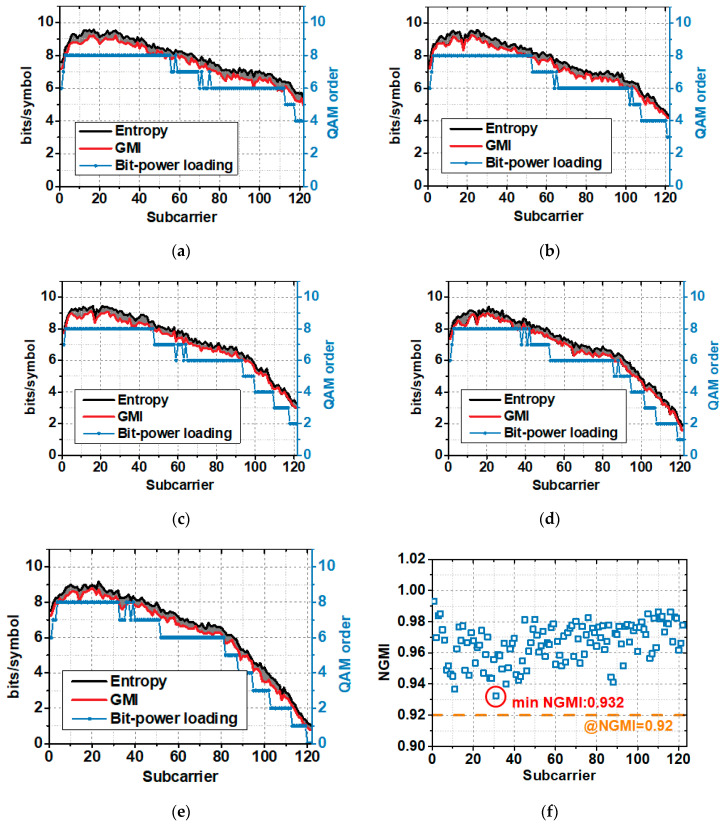
(**a**–**e**) Source entropy, GMI, and achievable bit loading QAM order versus subcarrier index at signal bandwidth from 1.3 GHz to 1.7 GHz; (**f**) NGMI versus subcarrier index at signal bandwidth of 1.5 GHz.

**Figure 9 sensors-20-06143-f009:**
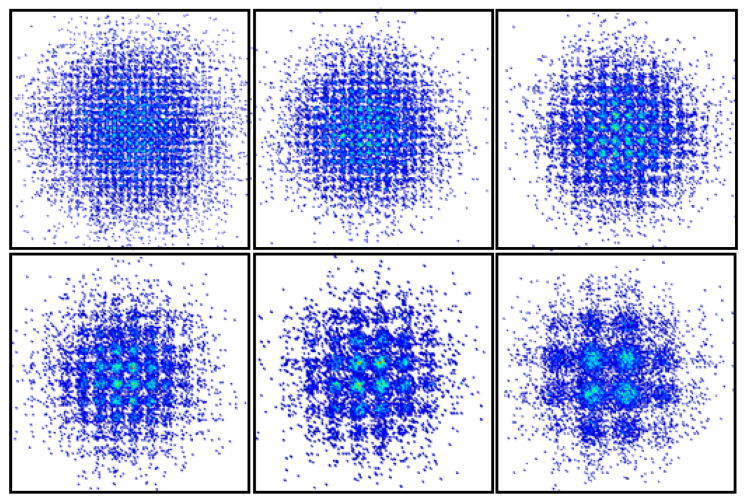
The corresponding constellation diagrams with different entropy.

**Table 1 sensors-20-06143-t001:** The GMI at different driven currents.

Current (mA)	37	47	57	67	77
GMI (bits/symbol)	5.2513	6.0569	6.8646	6.3487	5.8441

**Table 2 sensors-20-06143-t002:** The GMI at different signal Vpp.

Vpp (mV)	140	190	240	290	340
GMI (bits/symbol)	6.2112	6.8267	7.0334	6.7777	6.7004

**Table 3 sensors-20-06143-t003:** The AIR at different signal bandwidths.

Signal Bandwidth (GHz)	1.3	1.4	1.5	1.6	1.7
AIR (Gbps)	9.508	9.935	10.230	10.067	9.898
Min NGMI	0.931	0.933	0.932	0.926	0.921

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
