# Peer review of "Advanced Modulation Format of Probabilistic Shaping Bit Loading for 450-nm GaN Laser Diode based Visible Light Communication"

_sensors, 2020, doi:10.3390/s20216143_

Round 1

Reviewer 1 Report

This letter proposes an advanced modulation format based on probabilistic shaping (PS) bit loading envisioned for high data rate visible light communications.  The benefits of this modulation format are experimentally demonstrated using a hardware system based on a 450-nm Gallium Nitride laser diode. So, a 10.23 Gbps available information rate is successfully achieved at a signal bandwidth of 1.5GHz in a 1.2m free space transmission.

The letter is well organized and provides a convincing solution for VLC applications as it provides extensive experimental investigation that demonstrate the applicability and the performances of the proposed solution.

Section 1 provides a brief description of the VLC technology purpose – namely to provide high data rate communications using visible light - and enumerates the possible light sources that can be used in VLC emitters. Here, the authors suggest that LD are superior to LEDs as they have better light beam convergence and higher optical power.  Even if LDs have lower switching times compared to LEDs, the authors should be aware that the main purpose of a VLC system is to provide lighting, whereas the data transmission capacity should not affect in either way the lighting function (IEEE 802.15.7 standard). So, for the moment is quite difficult to used LD to provide lighting. In Section 1, the authors provide a short but comprehensive overview of current high data rate VLC systems based on LD, providing the motivation of the current work. The end of Section 1 emphasizes the novelty and the strong point of the work.

Section 2 describes the principles behind the proposed method, presenting the mathematical description of the model.

Section 3 presents the schematic of the VLC system and it illustrates the experimental setup. In the current configuration, the VLC receiver uses an optical filter to prevent the saturation of the APD. For practical utilization in mobile conditions, I consider that the VLC receiver should be designed based on a PIN PD instead of the APD together with an adaptive gain control circuit. This approach could reduce the chance of photoelement saturation and enable the usage in mobile conditions. In the current configuration, it is obvious that the VLC system is only suitable for laboratory testing, with the LD-based VLC emitter and the APD-based VLC receiver perfectly aligned.

Section 4 presents the results of the experimental evaluation. It provides a step by step evaluation in order to determine the optimal operating setting for the VLC system. I consider that the clarity of the presentation of this section can be improved.

Section 5 presents the conclusion of this work. This section is clear and adequately summarizes the results and the strong point of this work.

I consider that before publication, the authors should provide an adequate verification of the article in order eliminate some of the typos/errors. Please find some of them below:

“VLC has the potential to finish high-speed data” – a different verb could be used instead of finish.

“Then the PS bit loading signal is…” -  a comma is missing after “then”

“convent the light into electrical signal.” convent –> convert

However, form figure 4,…. form -> from

Figure 3 shows the luminance-voltage-current (L-V-I) characteristics of the LD and they were experimentally obtained at 24℃. -> which they were experimentally obtained at 24℃.

Avoid using “And” at the beginning of the sentence. Examples:

. And PS can be combined…

. And Experimental results show…

. And the system performance

The presentation of the letter could also be improved in terms of English usage and clarity of expression.

Additional discussions could be introduced in Section 4 in order to better present the benefits of the proposed method with respect to other works.

The authors presented only some of the benefits of the proposed method. Are there any downsides in terms of throughput, computation power required, complexity, or any other … ?

Reviewer 2 Report

This paper presents the experimental validation of a high-speed VLC system based on probabilistic shaping bit loading DMT modulation. The paper is clear, well organized and the numerical results support the potential of the proposed solution.

Main suggestions:

  • the introduction about the principles of probabilistic shaping could be enhanced.
  • Some comments about the expectations regarding the communication range and its impact on performance (the distance is set to 1.2 m in the experiments).
  • Some comments about the complexity and applicability of the proposed solution in practical VLC receivers.

Reviewer 3 Report

Some issues are expected to be more clear:

- In figure 2 the training data is from the received data. In real scenario, how can receiver feed back these information to transmitter.

- What is subcarrier in figure 5. In figure 2, there is not subcarrier on the optical signal. How subcarrier contribute to PS bit loading discrete multi-tone (DMT) modulation principle ? The performance evaluation discusses about it deeply.

- The distance link between transmitter and receiver is not considered in the section 2 analysis

- Temperature is mentioned but it did not contribute to the performance evaluation. Similarly with the LD frequency response.

Round 2

Reviewer 3 Report

Thanks for authors' contribution.